



# Integrating CVMix into GOTM (v6.0): A consistent framework for testing, comparing, and applying ocean mixing schemes

Qing Li[1], Jorn Bruggeman[2,3], Hans Burchard[4], Knut Klingbeil[4], Lars Umlauf[4], and Karsten Bolding[2,5]

[1]Fluid Dynamics and Solid Mechanics, Los Alamos National Laboratory, Los Alamos, New Mexico, USA
[2]Bolding & Bruggeman ApS., 5466 Asperup, Denmark
[3]Plymouth Marine Laboratory, Prospect Place, the Hoe, Plymouth PL1 3DH, UK
[4]Leibniz Institute for Baltic Sea Research Warnemünde, Rostock, Germany
[5]Department of Bioscience, Aarhus University, 8600 Silkeborg, Denmark

**Correspondence:** Qing Li (qingli@lanl.gov)

**Abstract.** The General Ocean Turbulence Model (GOTM) is a one-dimensional water column model including a set of state-of-the-art turbulence closure models, and has widely been used in various applications in the ocean modeling community. Here we extend GOTM to include a set of newly developed ocean surface vertical mixing parameterizations of Langmuir turbulence via coupling with the Community Vertical Mixing Project (CVMix). A Stokes drift module is also implemented in GOTM to provide the necessary ocean surface waves information to the Langmuir turbulence parameterizations, as well as to facilitate future development and evaluation of new Langmuir turbulence parameterizations. In addition, a streamlined workflow with Python and Jupyter Notebook is also described, enabled by the newly developed and more flexible configuration capability of GOTM. The newly implemented Langmuir turbulence parameterizations are evaluated against theoretical scalings and available observations in four test cases, including an idealized wind-driven entrainment case and three realistic cases at ocean station Papa, the northern North Sea and the central Gotland Sea, and compared with the existing General Length Scale scheme in GOTM. The results are consistent with previous studies. This development extends the capability of GOTM towards including the effects of ocean surface waves and provides useful toolsets for the ocean modeling community to further study the effects of Langmuir turbulence in a broader scope.

## 1 Introduction

The parameterization of vertical turbulent transport in the ocean surface boundary layer (OSBL) is an essential component of an ocean general circulation model (OGCM), representing the effects of unresolved small-scale boundary layer turbulence on redistributing heat, momentum and trace gases within the OSBL and mediating the exchange of these quantities between the atmosphere and the ocean interior. Li et al. (2019) compared a set of OSBL turbulent mixing parameterizations under idealized and realistic conditions, with a focus on those that include the effects of Langmuir turbulence. Significant discrepancies were



found among many OSBL parameterizations, both with and without Langmuir turbulence, highlighting the uncertainties in our understanding of the physical processes in the OSBL and the necessity of future development of a better OSBL turbulent mixing parameterization. While the comparison across many OSBL parameterizations in Li et al. (2019) provides useful insights into the relative importance of various different physical processes and forcings, a careful evaluation against observations is useful to assess the quantitative performance of these OSBL parameterizations.

The General Ocean Turbulence Model (GOTM, Burchard et al., 1999; Umlauf and Burchard, 2005; Umlauf et al., 2014, see updated version on gotm.net) is a one-dimensional water column model including a library of state-of-the-art turbulence closure models. It has been used for example to understand the evolution of thermal stratification in the North Sea and the northern Pacific (Burchard and Bolding, 2001), effects of breaking surface waves on surface boundary layer dynamics (see, e.g., Jones and Monismith, 2008), entrainment into bottom gravity currents (Arneborg et al., 2007), mixing in sloping bottom boundary layers (Umlauf and Burchard, 2010; Umlauf et al., 2015), and sediment dynamics (see, e.g., Conley et al., 2008; Burchard et al., 2013). Since GOTM has been coupled to biogeochemical models (Burchard et al., 2006; Bruggeman and Bolding, 2014), it has been used for several studies of marine ecosystems (see, e.g., Steiner et al., 2007; Hense and Quack, 2009; van der Molen et al., 2013; Kerimoglu et al., 2017; Powley et al., 2020). Furthermore, GOTM provides a useful platform both to compare across many OSBL parameterizations (e.g., Burchard and Bolding, 2001; Li et al., 2019), and to evaluate them against observations (e.g., Burchard et al., 2008). The turbulence module of GOTM has been integrated into several OGCMs, either by directly linking it as a module or by recoding, see, e.g., the General Estuarine Transport Model (GETM, Burchard and Bolding, 2002), the Regional Ocean Modelling System (ROMS, Haidvogel et al., 2000), the Nucleus for European Modelling of the Ocean (NEMO, Madec et al., 1991), the Semi-implicit Cross-scale Hydroscience Integrated System Model (SCHISM, Zhang et al., 2016), the Finite Volume Community Ocean Model (FVCOM, Chen et al., 2003) and the Proudman Oceanographic Laboratory Coastal Ocean Modelling System (POLCOMS, Holt and Umlauf, 2008).

The work of Li et al. (2019) assembles a set of new OSBL parameterizations with Langmuir turbulence using GOTM as the common driver. To make it available as part of a community model, a consistent implementation of the underlying algorithms has been made. A more general use of these OSBL parameterizations, e.g., in an OGCM, requires proper integration into the existing GOTM code. This paper documents the implementation of many functionalities described in Li et al. (2019) into the main GOTM repository by re-implementing some of the OSBL parameterizations in a way consistent with the existing GOTM code, promoting the coupling of the turbulence module to OGCMs. In particular, the efforts of incorporating the Community Vertical Mixing Project (CVMix, Griffies et al., 2015) in GOTM is described, which enables the K-profile parameterization (KPP, Large et al., 1994), as well as a few KPP variants that include the effects of Langmuir turbulence (e.g., Li et al., 2016; Reichl et al., 2016; Li and Fox-Kemper, 2017). A new Stokes drift module is also implemented in GOTM to facilitate the development and evaluation of Langmuir turbulence parameterizations in GOTM. While we are aware that Langmuir parameterizations have also been developed for second-moment turbulence models (e.g., Axell, 2002; Kantha and Clayson, 2004; Harcourt, 2013, 2015), we will implement and test these schemes in GOTM at a later point. The primary goal of this work is to extend the capability of GOTM, both as an ocean turbulence closure library and as a stand-alone one-dimensional water column model, to enable systematic comparison across many different parameterizations within a common framework. This





will allow us to precisely separate differences induced by variations in the turbulence parameterizations from those associated with different numerical schemes, parameterizations of the ocean surface fluxes, and other secondary effects.

In addition to documenting the implementation of CVMix in GOTM and the updates to improve the user interface, this paper also evaluates these new OSBL parameterizations in GOTM against available observations at a few sites. In particular,

we focus on the influence of Langmuir turbulence in such evaluations, which has never been done in GOTM.

The remainder of this paper is organized as follows. Development of incorporating CVMix and a Stokes drift module in GOTM is described in Section 2, together with an introduction of a streamlined workflow with Python (python.org) and Jupyter Notebook (jupyter.org) enabled by the newly developed and more flexible configuration capability in GOTM. Evaluation against available theories and observations in four test cases is presented in Section 3. This paper ends with a brief discussion

and main conclusions in Section 4.

## 2  Extending the functionality of GOTM

### 2.1  CVMix in GOTM

The Community Vertical Mixing Project (CVMix, Griffies et al., 2015) is a portable vertical mixing software package providing an extensible framework for the development of first-order turbulence closures. It provides a set of subroutines allowing

flexible implementation of surface and interior turbulence closures in the K-profile parameterization (KPP, Large et al., 1994; Van Roekel et al., 2018) in an ocean general circulation model (OGCM). These subroutines can be assembled in different ways to accommodate different needs (e.g., loop structure, available mean field variables) of the host OGCMs. Recently, a few Langmuir turbulence parameterization schemes based on KPP have also been included in CVMix (e.g., Li et al., 2016; Reichl et al., 2016; Li and Fox-Kemper, 2017). This allows easy implementations and testing of these Langmuir turbulence

parameterization schemes in other models.

The General Ocean Turbulence Model (GOTM, Umlauf and Burchard, 2005; Umlauf et al., 2014, see updated version on gotm.net) provides a collection of various turbulence closure schemes for the vertical mixing in the ocean and lakes, in particular second-order turbulence closures. The procedures and variables representing the implementation of these models are encapsulated in GOTM's FORTRAN module `turbulence` as illustrated in Fig. 1. This module can be easily integrated

in any existing library structure of a third-party OGCM or lake model. In addition, GOTM can also be used as a stand-alone one-dimensional water column model with flexible configurations to study the hydrodynamic and thermodynamic processes related to vertical mixing in natural waters. In this case, the turbulence routines are called from inside GOTM's main time loop implemented in the central FORTRAN module `gotm` (see Fig. 1). For these reasons, GOTM provides a useful platform for developing and comparing different ocean and lake vertical mixing parameterizations in both idealized and realistic scenarios

(e.g., Burchard and Bolding, 2001; Umlauf and Burchard, 2005; Burchard et al., 2008; Li et al., 2019).

As a first step towards extending the capability of GOTM to include Langmuir turbulence parameterizations, CVMix is incorporated in GOTM as an external library, including the above-mentioned first-order closure models of Langmuir turbulence based on KPP (see Fig. 1). Methodically, this is similar to the approach taken in Li et al. (2019) that enables the cross-





comparison among a set of Langmuir turbulence parameterization schemes in the single-column setup. Here, a user-interface
is developed and described that ensures consistency with the other modules of the GOTM code, such as the mean flow and
the meteorological forcing modules (interface module `gotm_cvmix`, see Fig. 1). Variables passed to CVMix through this
interface module include mean flow variables such as the currents, temperature and salinity, surface forcing variables such as
the surface friction velocity and surface buoyancy flux, and wave forcing variables such as the Langmuir number and Langmuir
enhancement factor. In return, turbulence variables such as the turbulent diffusivity and viscosity are passed back to the GOTM
main time loop.

Using the new interface to CVMix, it is now possible to directly call the CVMix subroutines, thereby making a range of
recent Langmuir turbulence parameterizations directly available in GOTM, and in OGCMs or lake models that use GOTM.
This allows to objectively compare state-of-the-art versions of the KPP model and second moment closure models. Note that
the CVMix interface is separate from the GOTM turbulence module (Fig. 1), so additional modifications to the source code of
the host model are still needed. However, such modifications are significantly less than would be needed if CVMix were to be
directly implemented in the host model.

## 2.2   Stokes drift in GOTM

Stokes drift (see a recent review by van den Bremer and Breivik, 2018, and references therein) is a key property of ocean
surface waves that is crucial for the dynamics and parameterizations of Langmuir turbulence (e.g., McWilliams et al., 1997;
Li et al., 2019). To provide the necessary information of the Stokes drift for various Langmuir turbulence parameterizations,
Stokes drift variables and a few options to configure the Stokes drift are implemented as a new module in GOTM. These
different options of Stokes drift provide a way to test the sensitivity of a Langmuir parameterization to the uncertainties in the
estimate of Stokes drift.

The most flexible option is to directly read in the Stokes drift profiles from a file (see FORTRAN module `stokes_drift`
in Fig. 1). The Stokes drift profiles can be either computed from the wave spectrum of direct measurements and wave models,
or estimated from some empirical relations. Tools for generating the Stokes drift input file for GOTM from various sources are
provided on Github (github.com/qingli411/gotmtool). To assist the development and testing of Langmuir turbulence parame-
terizations, two idealized options are also implemented (Fig. 1). The first option assumes a monochromatic surface wave, for
which the Stokes drift is an exponentially decaying profile with depth defined by the surface value $\boldsymbol{u}_0^S$ and a decay depth scale
$\delta^S$,

$$\boldsymbol{u}^S(z) = \boldsymbol{u}_0^S \exp\left(\frac{z}{\delta^S}\right), \tag{1}$$

where $z \leq 0$ is the water depth. The exponential profiles have been used in many idealized large eddy simulations of Langmuir
turbulence (e.g., McWilliams et al., 1997; Grant and Belcher, 2009). The second option assumes a Stokes drift profile that
depends only on the wind, derived from a set of empirical relations and assumptions (the "theory wave" approach of Li et al.,
2017, see their Eq. (25)). This "theory wave" approach estimates the Stokes drift profile from the wind assuming a $f^{-5}$ (where
$f$ is the frequency) spectral shape (sea also Breivik et al., 2016) with the directional spread correction of Webb and Fox-Kemper



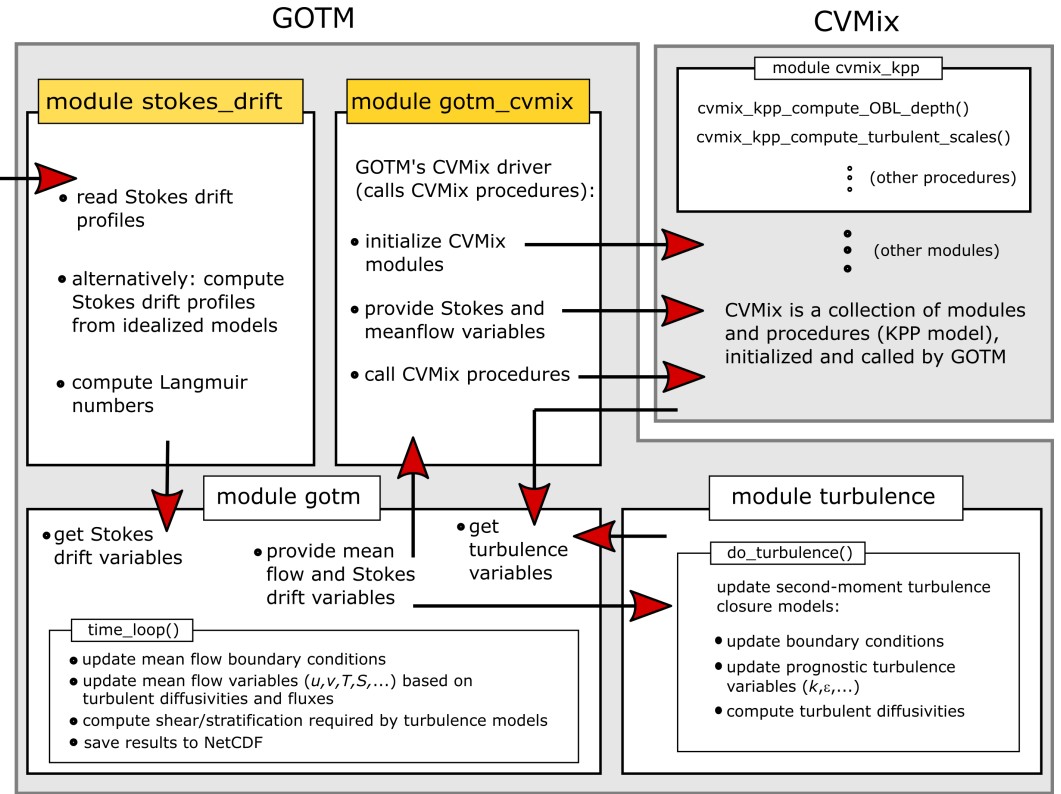

**Figure 1.** Schematic diagram illustrating the coupling of GOTM and CVMix. New GOTM modules are highlighted in yellow, interfaces between different model components are indicated as red arrows. Note that, for clarity, only a selection of the most relevant modules, procedures, and interfaces is shown.

(2015). The surface value and integrated transport of Stokes drift are estimated using empirical relations. By estimating the Stokes drift from the local wind, the contribution of swell is not explicitly represented, except a constant magnitude loss coefficient tuned against a WAVEWATCH III global wave hindcast simulation to represent the reduction effect by swell (Webb and Fox-Kemper, 2015). It has been shown to provide enough information of the ocean surface waves to allow a reasonable representation of the effects of Langmuir turbulence in OGCMs without coupling with a wave model through either a KPP variant (Li et al., 2017) or an energetics based planetary boundary layer scheme (Reichl and Hallberg, 2018; Reichl and Li, 2019). For both the exponential and the "theory wave" options, the controlling parameters (surface value and decay depth of Stokes drift in the former case and surface wind in the latter) can be set to constant or read from a file.

It should be noted that in this study the Stokes drift profile is only used in the Langmuir turbulence parameterizations without being integrated into the mean flow module in GOTM (e.g., Coriolis-Stokes force). This is consistent with the requirement of the two KPP-based Langmuir turbulence parameterizations used here. For other Langmuir turbulence parameterizations, direct





modifications of the mean flow equations in GOTM due to Stokes drift may be necessary (see, e.g., Appendix A of Li et al., 2019), which is straightforward as the full Stokes drift is now readily available in GOTM.

## 2.3 A streamlined workflow in GOTM

GOTM now supports the human-readable data-serialization language YAML (yaml.org) for the configuration of parameters. YAML is both more user-friendly and more developer-friendly than the FORTRAN namelist originally used in GOTM for the configuration. It has a clean and minimal syntax and is easy for extension and maintenance. The default values and the documentation of the configuration parameters of GOTM are now stored in the source code, so that the configurations can be generated from the compiled GOTM executable. This eliminates the need to save the default values of configuration parameters in a separate file, e.g., an XML file (w3.org/TR/xml11), as previously used in GOTM as well as many other ocean models. This is a useful feature especially for the development and maintenance of GOTM, since the configurations are always consistent with the source code. For GOTM users, this also guarantees that a compatible configuration file is always available whenever the source code is updated, which would require extra efforts if FORTRAN namelist files were used.

Taking advantage of this new feature of GOTM, a Python (python.org) wrapper of GOTM (i.e., an interface to access GOTM in Python) for easily working with GOTM in the Jupyter Notebook (jupyter.org) environment was also developed. After an inital setup step in the terminal, everything from building the GOTM executable, configuring the parameters, running GOTM simulations, to visualizing the simulation results can now be performed in the Jupyter Notebook environment. The source code and some examples are publicly available on Github (github.com/qingli411/gotmtool). The Jupyter notebooks to run the test cases and plot all the figures in this paper are available on Github (github.com/qingli411/A2020_CVMix_in_GOTM) for maximum reproducibility. They also serve as additional examples of using these tools.

## 3 Evaluation

We evaluate the newly implemented CVMix and Stokes drift modules in GOTM in three configurations: (i) KPP-CVMix: a typical KPP configuration with key parameters summarized in Table 1; (ii) KPPLT-VR12: a variant of KPP to account for the Langmuir turbulence enhanced mixing (Li et al., 2016); (iii) KPPLT-LF17: a variant of KPP to additionally account for the Langmuir turbulence induced entrainment (Li and Fox-Kemper, 2017). See Appendix A1 of Li et al. (2019) for a more detailed description of KPPLT-VR12 and KPPLT-LF17. The Generic Length Scale (GLS, Umlauf and Burchard, 2003) scheme in the $k$-$\varepsilon$ formulation with the weak-equilibrium stability function by Canuto et al. (2001), using a steady-state Richardson number of $Ri_{st} = 0.25$, denoted as GLS-C01A hereafter, is used as a reference. The parameters used in GLS-C01A are summarized in Table 2.

In the following sections, GOTM simulations with the above four vertical mixing schemes will be compared with available theoretical scalings or observations in four different test cases. We use a Cartesian coordinate system with $x$ and $y$ denoting the horizontal coordinates, $z$ the vertical (upward) coordinate, and $u$, $v$ and $w$ the corresponding components of the velocity.





**Table 1.** Summary of parameters and settings in KPP-CVMix.

| Description | CVMix parameter | Value / Reference |
|---|---|---|
| Critical Richardson number | `Ri_crit` | 0.3 |
| Nondimensional extent of surface layer | `surf_layer_ext` | 0.1 |
| Matching method between the OSBL and interior | `MatchTechnique` | `'SimpleShapes'` |
| Interpolation type for the bulk Richardson number | `interp_type` | `'quadratic'` |
| Interpolation type for the diffusivity and viscosity | `interp_type2` | `'LMD94'` |
| Enhance diffusivity at OSBL | `lenhanced_diff` | `.true.` |
| Limit the OSBL by the Ekman depth | `lEkman` | `.false.` |
| Limit the OSBL by the Monin-Obukhov length | `lMonOb` | `.false.` |
| Zero gradient of the shape function at OSBL | `lnoDGat1` | `.true.` |
| $C_v$ for the unresolved shear | `Cv` | Eq. (A3) of Danabasoglu et al. (2006) |
| Entrainment layer stratification | - | Eq. (39) of Van Roekel et al. (2018) |

Note that we are focusing on the OSBL mixing in KPP, and the interior mixing in CVMix is disabled here. OSBL: Ocean surface boundary layer.

**Table 2.** Summary of the generic length scale parameters in GLS-C01A.

| | $m$ | $n$ | $p$ | $c_1$ | $c_2$ | $c_3^-$ | $c_3^+$ | $\sigma_k$ | $\sigma_\psi$ | $\mathrm{Ri_{st}}$ |
|---|---|---|---|---|---|---|---|---|---|---|
| GLS-C01A | 1.5 | $-1.0$ | 3.0 | 1.44 | 1.92 | $-0.62^{\mathrm{a}}$ | 1.0 | 1.0 | 1.3 | 0.25 |

[a] Value computed from the steady-state Richardson number $\mathrm{Ri_{st}}$. See, e.g., Umlauf and Burchard (2003).

## 3.1 Idealized entrainment

165 The first test case is an idealized wind stress-driven entrainment case with no rotation, in which the OSBL gradually entrains into an underlying non-turbulent region with constant stable stratification. The GOTM simulation results can be directly compared with the relation derived from laboratory experiments (e.g., Price, 1979), in which the time evolution of the mixed layer depth $h_m$ follows

$$h_m(t) = (2R_v)^{1/4} u_* \left( \frac{t}{N_0} \right)^{1/2}, \tag{2}$$

170 where $R_v \approx 0.6$ is the bulk Richardson number, $u_*$ the water side surface friction velocity, and $N_0$ the initial buoyancy frequency. See more discussion on this relation and the model configuration in Umlauf and Burchard (2005).

We run the idealized entrainment case in GOTM for 30 hours on a vertical domain of 50 m with 250 equally spaced layers and a time step of 6 s. Here, we are using a relatively high vertical resolution and short time step to better capture the time evolution of the mixed layer. See Section 3.5 for a discussion of the sensitivity of the four vertical mixing schemes to the vertical

175 resolution and time step. The surface friction velocity is $u_* = 0.01 \text{ m s}^{-1}$ and the initial buoyancy frequency is $N_0 = 0.01 \text{ s}^{-1}$. Earth's rotation is not considered.



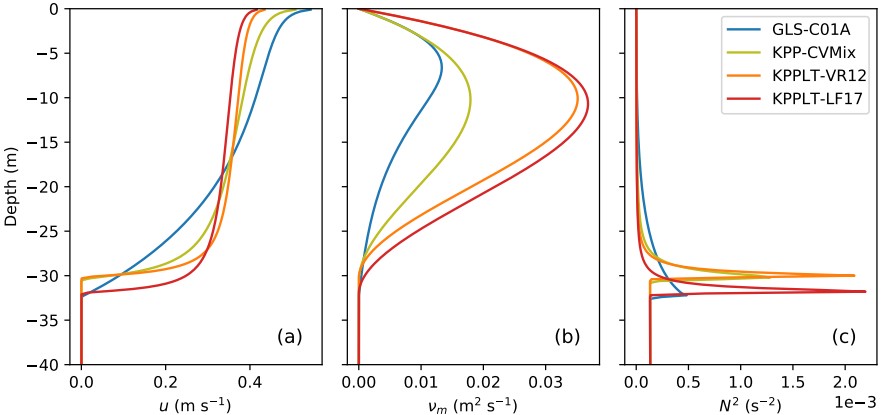

**Figure 2.** Vertical profiles of (a) down-wind velocity $u$, (b) turbulent viscosity $\nu_m$, and (c) squared buoyancy frequency $N^2$ at the end of the 30-hour simulations.

To test the effect of parameterizing the Langmuir turbulence enhanced mixing and the Langmuir turbulence enhanced entrainment in KPPLT-VR12 and KPPLT-LF17, we assume a Stokes drift in the wind direction (here in the $x$-direction) that exponentially decays with depth following Eq. (1), with a surface value of $u_0^S = |\boldsymbol{u}_0^S| = 0.11$ m s$^{-1}$ and decay scale of $\delta^S = 5$ m. This corresponds to a turbulent Langmuir number,

$$\mathrm{La}_t = \left(\frac{u_*}{u_0^S}\right)^{1/2} \approx 0.3, \tag{3}$$

at which Langmuir turbulence has a prominent influence on the turbulent mixing in the mixed layer (McWilliams et al., 1997).

Fig. 2 compares the vertical profiles of the velocity, $u$, turbulent viscosity, $\nu_m$, and squared buoyancy frequency, $N^2$, in all the four vertical mixing schemes at the end of the 30-hour simulations. KPP-CVMix predicts an enhanced turbulent viscosity than GLS-C01A, and the resulting velocity profile is more well-mixed in the mixed layer. KPPLT-VR12 accounts for Langmuir-enhanced mixing by enhancing the turbulent viscosity in KPP-CVMix. The resulting velocity profile is more well-mixed than KPP-CVMix. KPPLT-LF17 additionally accounts entrainment due to Langmuir turbulence which makes the mixed layer deeper. The turbulent viscosity is similar to KPPLT-VR12. These results are consistent with the design of KPPLT-VR12 and KPPLT-LF17. The strikingly smaller $N^2$ in the entrainment layer in GLS-C01A (panel c) is due to the downward diffusion of turbulent kinetic energy (TKE) into this layer, which is ignored in KPP.

Fig. 3 compares the time evolution of the mixed layer depth (MLD) in the four vertical mixing schemes with Eq. (2). Here the MLD is defined as the depth at which the squared buoyancy flux $N^2$ reaches its maximum (see Fig. 2c). An alternative definition based on a TKE threshold of $10^{-6}$ m$^2$ s$^{-2}$ as in Umlauf and Burchard (2005) yields similar results. Consistent with Umlauf and Burchard (2005), GLS-C01A matches the empirical law of the entrainment in Eq. (2) with excellent accuracy. The three KPP variants all over-predict the rate of deepening of the MLD at the beginning of the simulation and under-predict it at later times. As expected, KPPLT-LF17 predicts a deeper mixed layer depth than KPP-CVMix and KPPLT-VR12. Somewhat



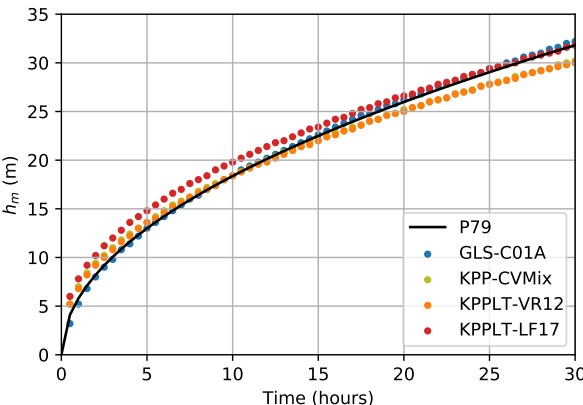

**Figure 3.** A comparison of the time evolution of the mixed layer depth $h_m$ in GOTM simulations defined by the depth at which $N^2$ reaches its maximum in color, and the prediction of Price (1979) in Eq. (2) in black.

counter-intuitively, KPPLT-VR12 predicts a very similar, or even slightly shallower, MLD as compared to KPP-CVMix in this case. This is because in KPPLT-VR12, the turbulent viscosity is enhanced and the down-wind velocity within the mixed layer is more well mixed and has weaker shear than in KPP-CVMix (Fig. 2). Since the boundary layer depth in KPP is determined by

the depth at which the bulk Richardson number first reaches a critical number (0.3 here), and the resolved shear term dominates the denominator of the bulk Richardson number (see Eq. (21) of Large et al., 1994) in this case, the reduced velocity shear in KPPLT-VR12 results in a slightly bigger bulk Richardson number and therefore a slightly shallower boundary layer depth than that in KPP-CVMix. This undesirable behavior of KPPLT-VR12 was one of the motivations to further include the Langmuir turbulence enhanced entrainment in KPPLT-LF17 (Li and Fox-Kemper, 2017).

## 3.2 Ocean Station Papa

The meteorological and oceanic observations at the Ocean Station Papa (OSPapa; 50.1°N, 144.9°W; pmel.noaa.gov/OCS/Papa) have been used to evaluate the performance of OSBL turbulent mixing schemes in many studies (e.g., Martin, 1985; Large et al., 1994; Kantha and Clayson, 1994; D'Alessio et al., 1998; Burchard and Bolding, 2001; Acreman and Jeffery, 2007), focusing mostly on the year 1961. Recent measurements of ocean surface waves at OSPapa (Thomson et al., 2013) allow us to evaluate

the effects of Langmuir turbulence parameterizations and assess the importance of Langmuir turbulence at this site.

  Here we use the temperature and salinity mooring data at OSPapa to initialize the GOTM simulations from rest in a 150 m vertical domain with 150 vertical grid cells. The time step is 60 s. Surface boundary conditions are set by the hourly surface fluxes data from March 21, 2012 to March 20, 2013. Throughout the year the Jerlov water type II (Paulson and Simpson, 1977) is assumed. Half-hourly wave spectral data collected using the Datawell Waverider buoy (cdip.ucsd.edu/metadata/166p1),

and binned into $n = 64$ frequency bands with $f_1 = 0.025$ Hz and $f_{64} = 0.58$ Hz, is used to estimate the Stokes drift for the Langmuir turbulence parameterizations. The Stokes drift profile is estimated from the band wave energy density spectrum $S_i$





according to:

$$\boldsymbol{u}^S(z) = \frac{16\pi^3}{g} \sum_{i=1}^{n} f_i^3 S_i \exp\left(\frac{8\pi^2 f_i^2 z}{g}\right) \hat{\boldsymbol{e}}_i^W \Delta f_i, \tag{4}$$

where $f_i$ is the band center frequency, $\Delta f_i$ the bandwidth, $\hat{\boldsymbol{e}}_i^W$ a unit vector in the band mean direction and $g$ the gravity

acceleration. The grid cell averaged value is computed following Appendix B of Harcourt and D'Asaro (2008). For simplicity, we are ignoring the effect of wave spreading, which may lead to an overestimation of the Stokes drift (e.g., Webb and Fox-Kemper, 2015). Note that unlike Li et al. (2019), we are not including the contribution of a $f^{-5}$ spectral tail beyond the cutoff frequency. A spectral tail contributes more to the surface value of Stokes drift, but much less to the surface layer (here upper 20% of the mixed layer) averaged Stokes drift, which is used in the Langmuir turbulence parameterizations here.

The annually-averaged net heat flux and freshwater flux over this one-year period are 31.7 W m$^{-2}$ and 12.9 mg m$^{-2}$ s$^{-1}$, respectively. Such imbalance of the heat and freshwater fluxes would increase the temperature of a water column of 100 m by about 2.5 °C over a year, and decrease the salinity by about 0.1 g kg$^{-1}$.[1] To directly compare with the temperature and salinity measurements in single-column simulations over a long period, such imbalance in the heat and freshwater fluxes needs to be compensated by careful adjustments to account for the effects of vertical advection and lateral processes (e.g., Large,

1996). Our focus here is to show the effects of parameterizing Langmuir turbulence in GOTM. Therefore, instead of trying to balance the heat and freshwater budget in a rather empirical way, we break the seasonal cycle into four relatively shorter stages (see Fig. 4). These four stages roughly represent (I) the spring restratification, (II) stable forcing in summer, (III) mixed layer entrainment in fall and winter, and (IV) preconditioning for restratification in winter, each of the stages being initialized by observed temperature and salinity profiles. In this way, the differences between different vertical mixing schemes can be

shown under different forcing regimes, using the observation as a reference, and the accumulative effects of the ignored vertical advection and lateral processes are reduced.

As shown in Fig. 4e, all four vertical mixing schemes predict warmer SST than the observation throughout the year, especially in stages II and III, and saltier SSS in stage III and slightly fresher SST in stage IV. This is likely a result of missing the vertical advection and lateral processes. Correspondingly, the MLD in all four vertical mixing schemes except KPPLT-LF17 is

mostly shallower than the observation throughout the year, especially in stage III and during sporadic mixed layer deepening events in stage I and IV (Fig. 4g). As expected, incrementally including the effects of Langmuir turbulence enhanced mixing and entrainment in KPPLT-VR12 and KPPLT-LF17 makes the MLD deeper and SST cooler. KPPLT-LF17 appears to match the observation the best. We note that this is not sufficient evidence of KPPLT-LF17 outperforming KPPLT-VR12 and KPP-CVMix without carefully accounting for the effects of vertical advection and lateral processes. However, this result suggests that the

effects of Langmuir turbulence on SST and MLD are significant and can be comparable to the effects of vertical advection and lateral processes at OSPapa.

Fig. 5 shows the temperature and salinity profiles at the end of each stage at OSPapa. All four vertical mixing schemes give too sharp temperature and salinity gradients at the base of the mixed layer than the observation, and, especially in stages I and II, too strong temperature gradient within the mixed layer. Again, we can see that KPPLT-LF17 appears to match the

---

[1]In this estimate we assume the seawater has a density of 1026 kg m$^{-3}$, a specific heat of 3985 J kg$^{-1}$ °C$^{-1}$ and a salinity of 33 g kg$^{-1}$.



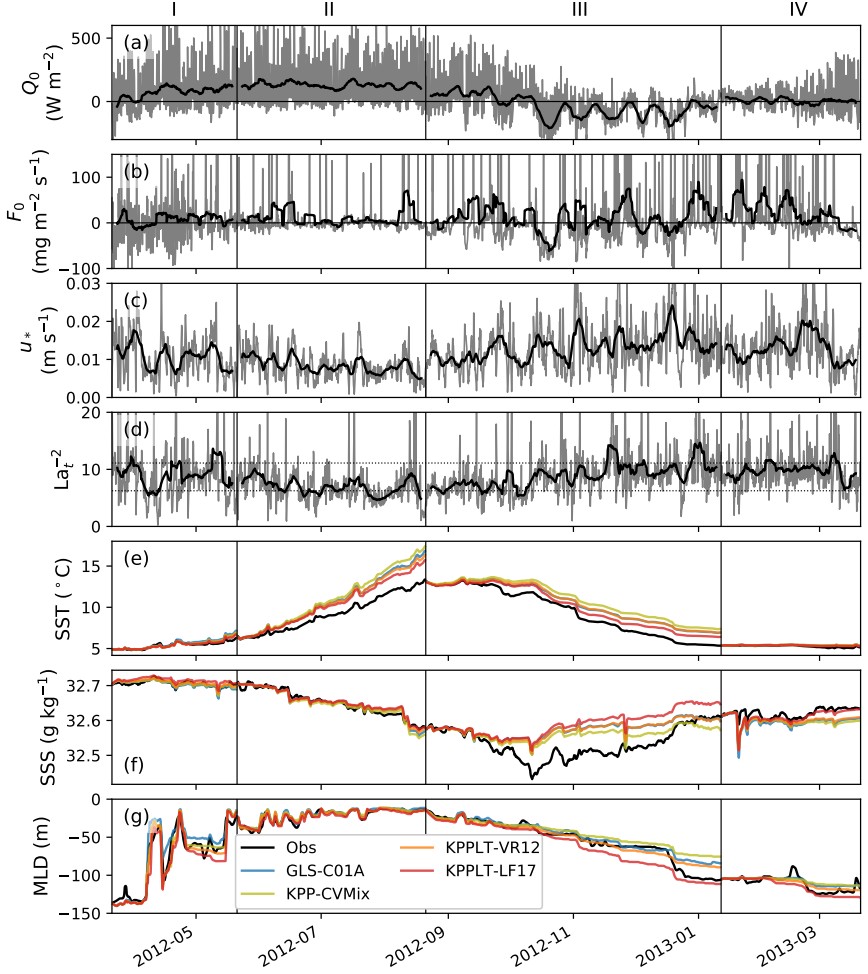

**Figure 4.** Time series of (a) net surface heat flux (W m$^{-2}$), (b) net freshwater flux (precipitation minus evaporation; mg m$^{-2}$ s$^{-1}$), (c) surface friction velocity (m s$^{-1}$), (d) La$_t^{-2}$ where La$_t$ is the turbulent Langmuir number, (e) sea surface temperature (SST; °C), (f) sea surface salinity (SSS; g kg$^{-1}$) and (g) mixed layer depth (MLD; m) defined by a temperature threshold method following de Boyer Montégut et al. (2004), at the Ocean Station Papa. In (a)-(d), the thin line in gray shows the 3-hourly time series and the thick line in black shows the 5-day moving average. The black dotted lines in (d) marks the values corresponding to La$_t$ = 0.3 and La$_t$ = 0.4. In (e)-(g), the black line shows the daily-averaged measurements at the Ocean Station Papa. Daily-averaged simulation results of GLS-C01A, KPP-CVMix, KPPLT-VR12 and KPPLT-LF17 are shown in blue, yellow, orange and red, respectively. In each panel, the one-year time series is composed of time series of four stages, chosen to roughly represent the conditions of (I) spring restratification, (II) stable forcing, (III) mixed layer entrainment and (IV) preconditioning for restratification, respectively.

observed profiles the best, while KPP-CVMix performs the worst. The differences in the shape of the observed and simulated profiles, especially the mismatch below the mixed layer, suggests the importance of other processes than vertical mixing that

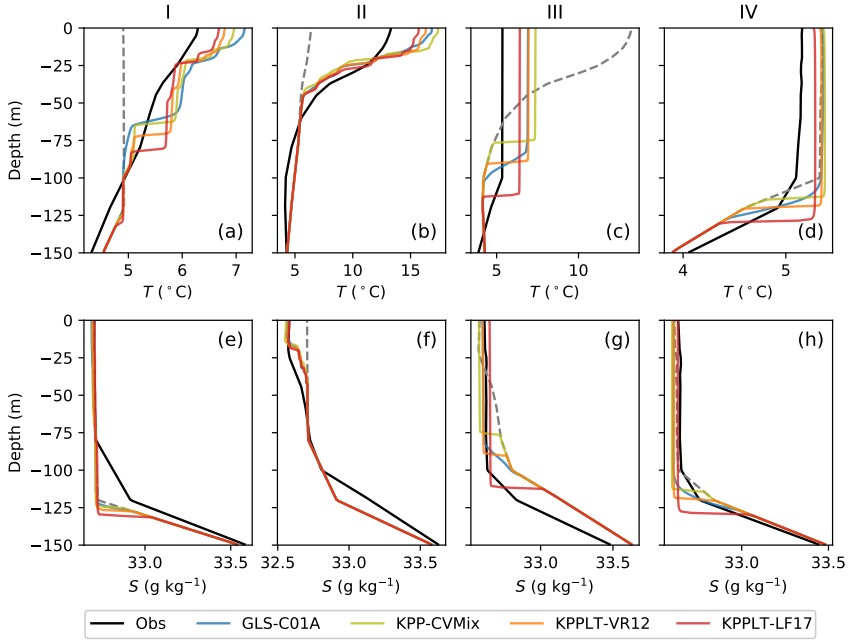

**Figure 5.** A comparison of the vertical profiles of the daily mean temperature (panels (a)-(d)) and salinity (panels (e)-(h)) at Ocean Station Papa between the observation (black) and GOTM simulations (colored). The four columns show the profiles at the end of the four stages shown in Fig 4. Dashed line in gray in each panel shows the initial condition at the beginning of each stage.

are not included in the single column GOTM simulations here. It is interesting to note that KPP based parameterizations all give sharper density interface at the OSBL base than GLS-C01A. Again, this may be due to the downward diffusion of TKE into the interface in GLS-C01A, which is missing in KPP.

Fig. 6 highlights the effects of parameterizing Langmuir turbulence by comparing the simulated temperature evolution in GOTM between the four parameterizations. Consistent with Li et al. (2019), the effects of Langmuir turbulence in KPPLT-VR12 and KPPLT-LF17 are strongest during mixed layer deepening in the fall (stage III) and the sporadic mixing events when the mixed layer is shallow (stage II). Its effects are weaker when the mixed layer is deep in winter (stage I and IV), even though both the winds and waves are stronger in winter (Fig. 4c,d). Such effects of Langmuir turbulence in KPP-based

parameterizations are significant as compared to the difference between KPP-CVMix and GLS-C01A.

    To test the sensitivity of the Langmuir turbulence parameterizations to the uncertainties in estimating the Stokes drift, we repeat the GOTM simulations with Stokes drift estimated from (i) the "theory wave" approach in Li et al. (2017), and (ii) an idealized exponential profile assuming $\delta^S = 5$ m and $\mathrm{La}_t = 0.3$ in Eqs. (1) and (3). The "theory wave" estimate of Stokes drift is often an underestimate as the effects of swell are largely ignored. The exponential profile, which represents an idealized

swell (but in the direction of the local wind), is likely an overestimation of the Stokes drift for most cases, especially given that the Stokes drift in the real ocean typically decay much faster than exponential (Webb and Fox-Kemper, 2011, 2015). This

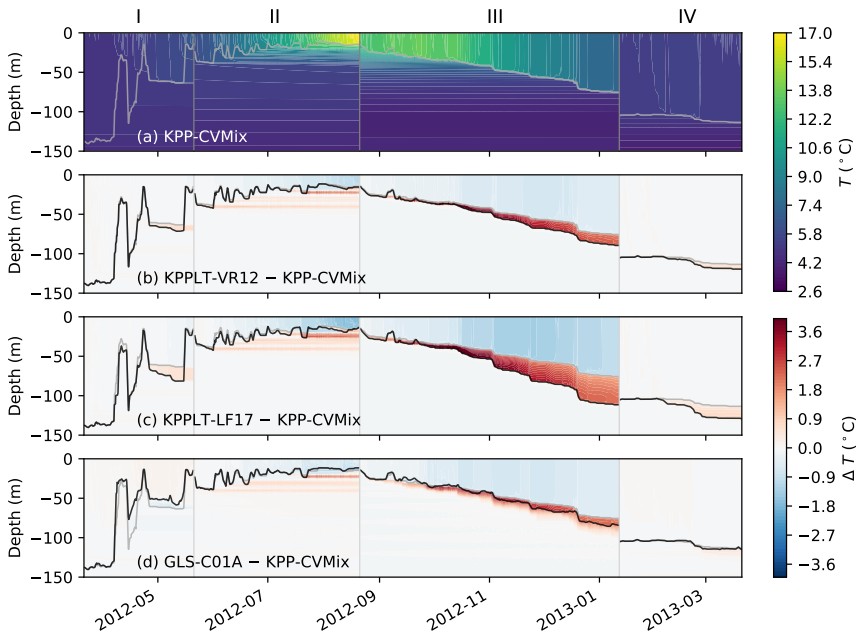

**Figure 6.** Hovmöller diagrams comparing the simulated temperature between the four vertical mixing schemes at Ocean Station Papa. (a) shows the simulated temperature in KPP-CVMix. (b)-(d) show the differences from KPP-CVMix in KPPLT-VR12, KPPLT-LF17 and GLS-C01A, with the black line showing the mixed layer depth (MLD). In all panels the gray line shows the MLD in KPP-CVMix as a reference. KPP-CVMix is used as the reference here simply for better visualization of the effects of Langmuir turbulence parameterizations (panels b and c).

is indeed the case for OSPapa, as shown in Fig. 7 which compares the distributions of the Stokes drift profiles estimated from the "theory wave" approach (blue) and an exponential profile (green) with that computed from the observed wave spectrum (black). Here the Stokes drift profiles are normalized by the surface friction velocity as Stokes drift tends to vary with the wind

(see Fig. 4c,d). Note that there is a seasonal variation of the relation between Stokes drift and the wind at OSPapa that is better captured by the "theory wave" estimate but not in the exponential profile estimate (stage II).

Figs. 8a,b compare the time series of $\mathrm{La}_t^{-2}$ and $\mathrm{La}_{\mathrm{SL}}^{-2}$ from the two estimates of Stokes drift with that computed from the observed wave spectrum, where $\mathrm{La}_{\mathrm{SL}}$ is the surface layer averaged Langmuir number (Harcourt and D'Asaro, 2008),

$$\mathrm{La}_{\mathrm{SL}} = \left( \frac{u_*}{\langle u^S \rangle_{\mathrm{SL}} - u_{\mathrm{ref}}^S} \right)^{1/2}, \tag{5}$$

in which $\langle u^S \rangle_{\mathrm{SL}}$ is the surface layer averaged Stokes drift and $u_{\mathrm{ref}}^S$ is a reference Stokes drift at the base of the mixed layer. The "theory wave" approach of Li et al. (2017) gives a slightly smaller estimate of the Stokes drift and the exponential profile approach gives a much bigger estimate for most of the times, especially for the surface layer averaged values. Panels (c)-(e) compare the simulated SST, SSS and MLD between the three simulations. It is seen that KPPLT-VR12 and KPPLT-LF17 give

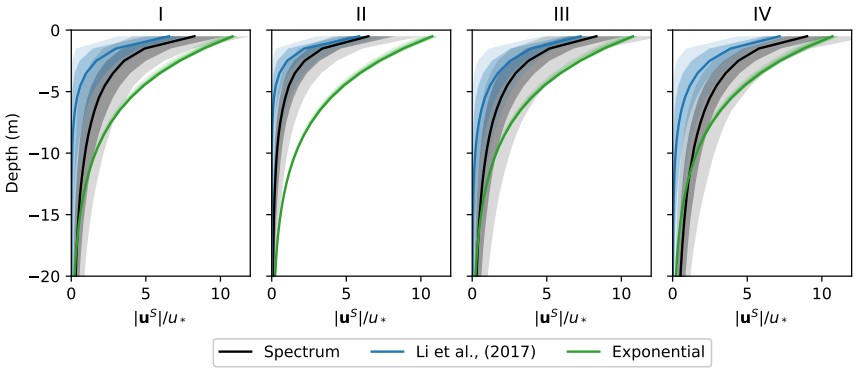

**Figure 7.** A comparison of the Stokes drift profiles estimated from the observed wave spectrum in black, "theory wave" approach of Li et al. (2017) in blue and an idealized exponential profile with $\mathrm{La}_t = 0.3$ and $\delta^S = 5$ m in green during the four stages of a seasonal cycle at the Ocean Station Papa. Stokes drift is normalizd by the surface friction velocity $u_*$. The lines show the median, and the dark and light shadings enclose 50% (25th to 75th percentiles) and 80% (10th to 90th percentiles) occurrence during each stage.

very similar results with both the two estimates of Stokes drift and that computed from the observed wave spectrum, except the

SST in stage II where the exponential profile approach significantly overestimates the Stokes drift, and therefore Langmuir-enhanced mixing and near-surface cooling. This is consistent with the findings of Li et al. (2017), suggesting that we may use the "theory wave" estimate of Stokes drift for KPPLT-VR12 and KPPLT-LF17 in GOTM simulations for cases where sufficient wave measurements are not available, such as for the two cases discussed in the following sections. It should be noted that both KPPLT-VR12 and KPPLT-LF17 use only the Langmuir numbers to parameterize the effects of Langmuir turbulence, which are

relatively insensitive to the exact profile of Stokes drift. For vertical mixing schemes that depends on the full profile of Stokes drift (e.g., Harcourt, 2013, 2015), this "theory wave" estimate might not be sufficient.

## 3.3  FLEX

The Fladen Ground Experiment (FLEX) test case is based on an intensive field campaign carried out in spring 1976 in the northern North Sea at 58°55'N and 0°32'E at a depth of about 145 m. Between April 6 and June 13 regular CTD (conductivity-

temperature-depth) profiles were sampled that were compiled by Soetje and Huber (1980) into vertical profiles of potential temperature and salinity. These profiles show the transition from fully mixed to stratified conditions in the upper half of the water column with a top to mid-depth temperature difference of 4 K. The salinity stratification remains weak with a maximum bottom to top difference of 0.1 g kg$^{-1}$. Bottom-generated turbulence due to weak tidal currents keeps the lower half of the water column mixed and supports deepening of the thermocline. The meteorological forcing was highly variable including

several storms that led to intermittent mixed layer deepening. Ship-based meteorological data for wind speed, dry and wet air temperature, air pressure, short-wave radiation and long-wave back radiation are available. Since lateral advection is weak in this region and the development of thermal stratification depends on a subtle balance of stratifying forces of surface warming





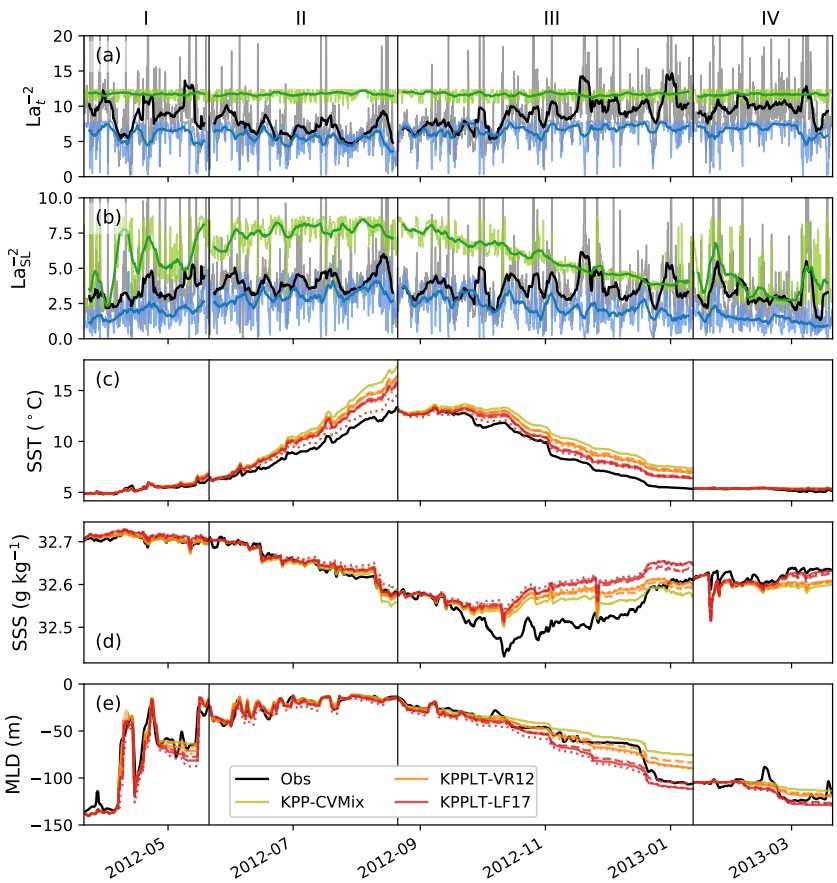

**Figure 8.** Time series of (a) $\mathrm{La}_t^{-2}$ where $\mathrm{La}_t$ is the turbulent Langmuir number, (b) $\mathrm{La}_{\mathrm{SL}}^{-2}$ where $\mathrm{La}_{\mathrm{SL}}$ is the surface layer averaged Langmuir number, (c) sea surface temperature (SST; °C), (d) sea surface salinity (SSS; g kg$^{-1}$) and (e) mixed layer depth (MLD; m) defined by a temperature threshold method following de Boyer Montégut et al. (2004), at the Ocean Station Papa. In (a) and (b), the thin line in gray shows the 3-hourly time series and the thick line in black shows the 5-day moving average estimated from the wave spectrum using (4). The blue lines show the same for the "theory wave" estimate of Stokes drift following Li et al. (2017) and green lines for the exponentially decaying Stokes drift assuming $\mathrm{La}_t \approx 0.3$ and $\delta^S = 5$ m. Solid lines in (c)-(e) show the same results as in panels (e)-(g) of Fig. 4. Dashed lines show the results with Stokes drift estimated from the "theory wave" approach. Dotted lines show the results with the idealized exponential Stokes drift.

and de-stratifying forces of wind and tidal mixing, the FLEX data set has become a standard test case for surface mixed layer models (e.g., Friedrich, 1983; Frey, 1991; Burchard and Baumert, 1995; Burchard et al., 2006) including GOTM.

Since the focus of the present model development is on the surface boundary layer and CVMix does not contain a bottom boundary layer module at the moment, we simulate the FLEX test case with our four vertical mixing schemes without tidal





forcing. However, to illustrate the relative effect of Langmuir turbulence versus tidal forcing, we also run GLS-C01A with tidal forcing as a reference.

Here all five GOTM simulations are initialized with temperature and salinity profiles from April 6, 1976, and run through
Jun 7, 1976, forced by hourly meteorological data. The surface heat flux is computed internally in GOTM from the meteorological data following Fairall et al. (1996). The surface freshwater flux is ignored. The local depth is 145 m, resolved with 145 evenly distributed grid cells. A time step of 360 s is used, and 3-hourly output is analyzed here. Since stratification is dominated by temperature in this case, the salinity field is relaxed towards the observations on a time scale of 48 hours, and we focus our discussion on the temperature field.

There are no direct measurements of ocean surface waves in the FLEX case, therefore we are using the "theory wave" approximation of Li et al. (2017) to estimate the Stokes drift from the wind speed, assuming that wind and waves are aligned. As demonstrated in the previous section, this approximation provides a reasonable estimate of the Stokes drift in our Langmuir turbulence parameterizations KPPLT-VR12 and KPPLT-LF17.

Consistent with previous studies, GLS-C01A reproduces the observed SST and MLD with good accuracy (blue versus black
lines in Fig. 9d,e). The tidal forcing has very small impact on the simulated SST and MLD (dashed versus solid lines in blue), providing some support for neglecting the tides in following discussion of the KPP simulations (see above). The SSTs predicted by the three variants of KPP are generally close to that computed by GLS-C01A, with progressively cooler SST the more effects of Langmuir turbulence are accounted for. Accordingly, the MLD is the deepest in KPPLT-LF17 and shallowest in KPP-CVMix. Similar to the stage I of the OSPapa case (see Fig. 4g), the effect of Langmuir-induced entrainment in KPPLT-
LF17 is most effective in the sporadic mixed layer deepening events, especially around May 12 (Julian day 133) when a storm passed across the site (e.g., Burchard and Baumert, 1995).

Fig. 10 compares the simulated temperature distribution with the observations. To highlight the effects of tidal forcing and Langmuir turbulence, panel (c) shows the difference in GLS-C01A with and without tidal forcing, and panel (e) and (f) show the differences in KPP with and without Langmuir turbulence. Both GLS-C01A and KPP-CVMix appear to reproduce
the dominant features in the evolution of the temperature profile. Specifically, stratification gradually develops, with sporadic mixing events until around May 12 when a storm passed and mixed the water in the upper about 50 m. Afterwards, a second surface warm layer of about 20 m develops on top of the deeper mixed layer. KPP-CVMix gives slightly shallower initial restratification than GLS-C01A from around April 22 to May 6 (see the white solid lines in panels (b) and (d)). The effects of Langmuir turbulence in deepening the mixed layer is apparent during this time (see panels (e) and (d)), but more significant
when the storm passed around May 12, resulting in cooler temperature within the mixed layer and warmer at the base of the mixed layer as compared to KPP-CVMix. Such effects are comparable in magnitude to the effect of tidal forcing. As shown in panel (c), the inclusion of tides increases the mixing across the thermocline, and thus across the entire water column, which is reflected in a cooling of the mixed layer and a warming of the underlying layers. The tidal mixing across the thermocline will likely redistribute the Langmuir-induced warming at the mixed layer base down to deeper layers, resulting in even stronger
cooling of the mixed layer and warming underneath. A second-moment closure scheme that includes the effects of Langmuir

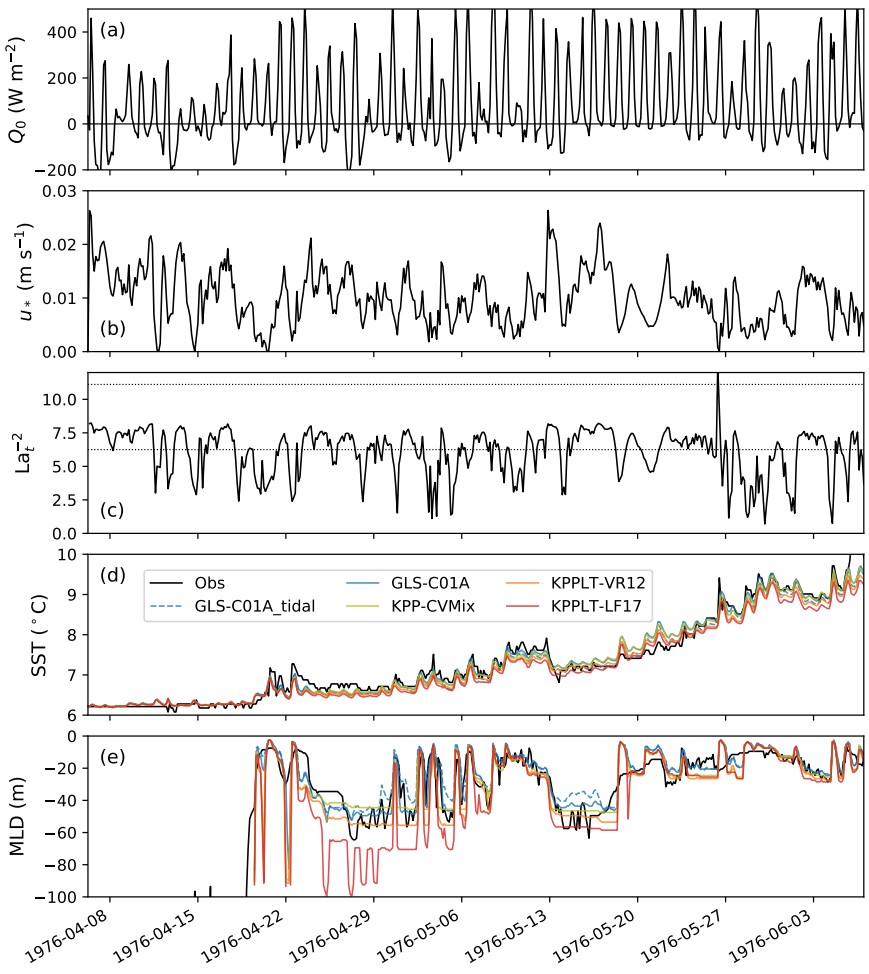

**Figure 9.** Three-hourly time series of (a) net surface heat flux (W m$^{-2}$), (b) surface friction velocity (m s$^{-1}$), (c) La$_t^{-2}$ where La$_t$ is the turbulent Langmuir number, (d) sea surface temperature (SST; °C), and (e) mixed layer depth (MLD; m) defined by a 0.2 °C temperature threshold referenced to the surface, in the FLEX case. The black dotted lines in (c) marks the values corresponding to La$_t = 0.3$ and La$_t = 0.4$. In (d) and (e), the black line shows the observations and colored lines in blue, yellow, orange and red show the simulation results of GLS-C01A, KPP-CVMix, KPPLT-VR12 and KPPLT-LF17, respectively. Results of GLS-C01A with tidal forcing are also shown for reference (dashed lines in blue).

turbulence (e.g., Harcourt, 2013, 2015) will be helpful to directly quantify the combined effects of Langmuir turbulence and tides.



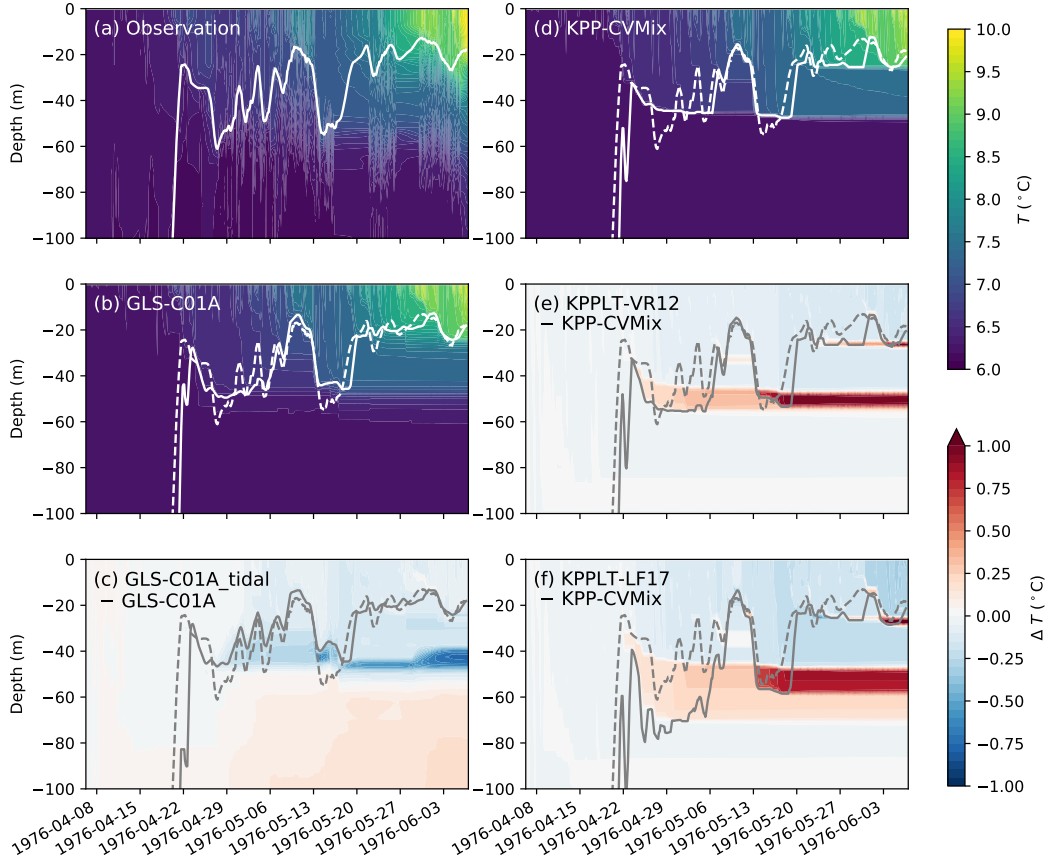

**Figure 10.** Hovmöller diagrams comparing the observed and simulated temperature between the four vertical mixing schemes in the FLEX case. (a) shows the observed temperature. (b) shows the simulated temperature in GLS-C01A. (c) illustrates the effect of tidal forcing by showing the difference between GLS-C01A with and without tides. (d) shows the simulated temperature in KPP-CVMix. (e) and (f) show the differences from KPP-CVMix in KPPLT-VR12 and KPPLT-LF17. Solid lines show the mixed layer depth (MLD; m) in each case defined by a 0.2 °C temperature threshold referenced to the surface. For reference the observed MLD is also shown in panels (b)-(f) in dashed lines.

### 3.4 Gotland Basin

Finally, we tested the four vertical turbulent mixing schemes in the central eastern Gotland Basin of the Baltic Sea (57.3°N, 20.0°E). The primary goal of this analysis is to test the performance of these schemes in a multi-year simulation of a non-tidal basin that remains stratified throughout the year due to a permanent halocline (Feistel et al., 2008). A detailed description of the GOTM setup for the Gotland Basin is described in Burchard et al. (2006).

Here, we used the hourly meteorological forcing from the COSMO-REA6 regional reanalysis data for Continental Europe (Bollmeyer et al., 2015). The GOTM simulations were carried out for the years 1997-2002, during which the mean net heat flux



from this data set was -1.1 W m$^{-2}$. A vertical domain of 250 m (corresponding to the deepest point of the basin) was discretized

with 250 evenly spaced vertical layers and a time step of 10 minutes. Daily output was saved for the analysis. Salinity was

nudged to observations, available at approximately 3 months resolution from a governmental monitoring program, with a

relaxation time scale of 50 days. Similar to the FLEX test case, we are focusing our discussion on the temperature field. The

E.U. Copernicus Marine Environment Monitoring Service (CMEMS) wave hindcast simulation of the Baltic Sea[2], forced by

the ECMWF's ERA5 reanalysis products, was used to provide the hourly data for the wave variables in the Gotland Basin. The

simulated significant wave height in the Baltic Sea wave model product is validated against the observations at the *Northern*

*Baltic* wave buoy[3] in the northern part of the Central Baltic Sea (not shown).

As for this wave product only the surface Stokes drift was available, we estimated the Stokes drift profiles from an approach

similar to the "theory wave" estimate of Li et al. (2017). These authors approximate the vertical profile of the Stokes drift by

assuming a wave spectrum proportional to $f^{-5}$ (where $f$ is the frequency) and the directional spreading correction of Webb

and Fox-Kemper (2015), subject to constraints on the peak frequency, the surface Stokes drift and the vertically integrated

Stokes drift, all estimated from the wind speed using empirical relations. Here, we assume the same vertical profile shape, but

replace the empirical constraints by available data from the wave hindcast simulation. Similar approaches to estimate the full

Stokes drift profiles from standard wave model output are also discussed in, e.g., Breivik et al. (2016).

Fig. 11a-c shows the time series of the forcing conditions; a comparison of the simulated SST and MLD to the observations

in the Gotland Basin is provided in Fig. 11d,e. Consistent with previous test cases, KPP-CVMix gives weaker fall-to-winter

mixed layer deepening than GLS-C01A. Progressively accounting for additional effects related to Langmuir turbulence in

KPPLT-VR12 and KPPLT-LF17 increases the MLD by up to around 15 meters in the late winter. However, this doesn't seem

to significantly change the simulated SST. The simulated SSTs in the four simulations are hardly distinguishable from each

other at this multi-year time scale, and are consistent with the available observations.

Vertical profiles of the simulated temperature in these simulations are compared with the observations at selected dates in

Fig. 12. The dates (also marked by gray diamonds in Fig. 11) are selected to roughly represent two seasonal cycles. For some

of the observations, multiple measurements spanning a period of a few days were recorded (cluster of plus signs in Fig. 11d,e).

It is interesting to note that the SST and MLD estimated from the observed temperature profiles can be very different inside

these clusters, suggesting variations due to processes with relatively fast time scale. For this reason, we also show examples of

these different temperature profiles in Fig. 12b,d (dotted lines). The most important observation is that the mixing schemes that

include Langmuir effects result in significantly increased MLDs in late fall and early winter, when mixed-layer deepening is

strongest (Fig. 12d,h). As the thermocline has already been completely eroded during these periods, the mixed layer is bounded

from below by the permanent halocline of the Gotland Basin, typically located around 50-70 m depth (not sown). This suggest

that Langmuir effects significantly contribute to the halocline erosion in winter, which is especially relevant from an ecosystem

modeling perspective as the halocline is known to separate the nutrient-depleted surface waters from the nutrient-rich deeper

layers (Feistel et al., 2008).

---

[2]https://resources.marine.copernicus.eu/?option=com_csw&view=details&product_id=BALTICSEA_REANALYSIS_WAV_003_015

[3]https://www.emodnet-physics.eu/map/platinfo/PIROOSDownload.aspx?PlatformID=8793



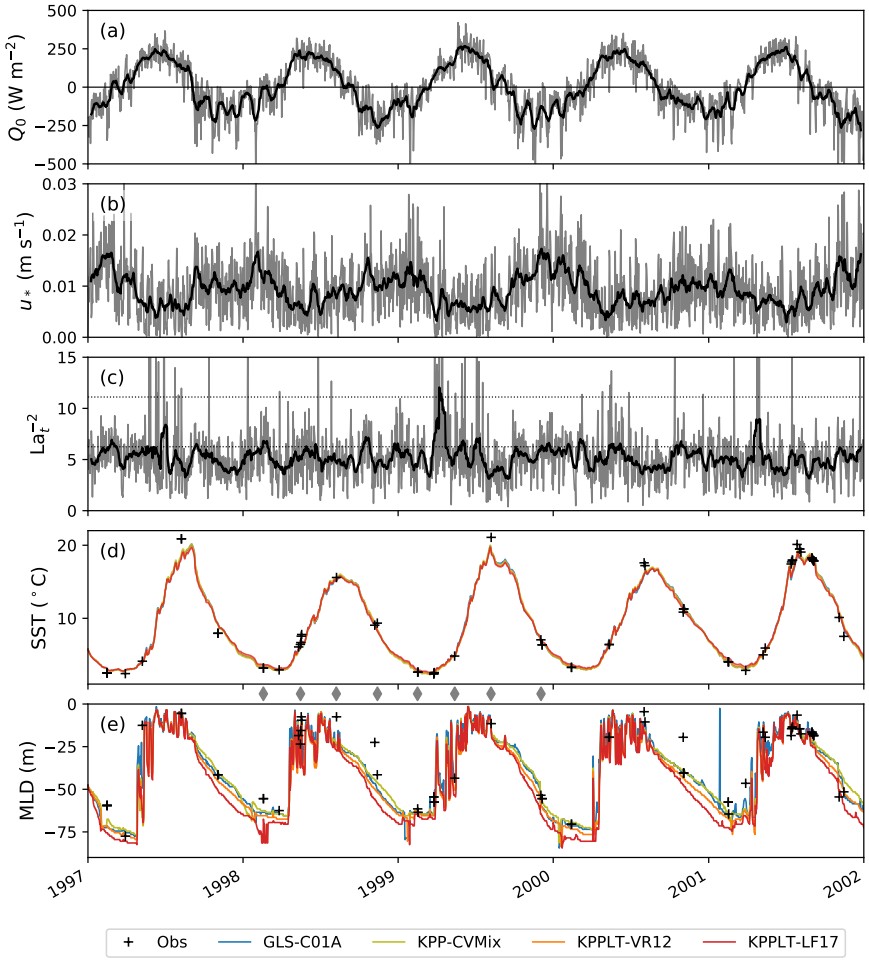

**Figure 11.** Five-year time series of (a) net surface heat flux (W m$^{-2}$), (b) surface friction velocity (m s$^{-1}$), (c) La$_t^{-2}$ where La$_t$ is the turbulent Langmuir number, (d) sea surface temperature (SST; °C), and (e) mixed layer depth (MLD; m) defined by a 0.2 °C temperature threshold referenced to the surface, at the Gotland. In (a)-(c), the thin line in gray shows the daily time series and the thick line in black shows the 15-day moving average. The black dotted lines in (c) marks the values corresponding to La$_t$ = 0.3 and La$_t$ = 0.4. In (d) and (e), the black plus signs show the observations at Gotland. Daily results of GLS-C01A, KPP-CVMix, KPPLT-VR12 and KPPLT-LF17 are shown in blue, yellow, orange and red, respectively. Gray diamonds between (d) and (e) mark the time when the vertical profiles of temperature are compared with the observations in Fig. 12.

For periods outside the winter months, however, all four vertical mixing schemes give quite similar results, especially during spring (Fig. 12b,f) and summer (Fig. 12c,g). The difference from the observed temperature profiles may be related to processes missing in the single column model used in our study, for example those associated with the basin-scale doming of the density structure (see, e.g., Fig. 9 in Holtermann et al., 2014)

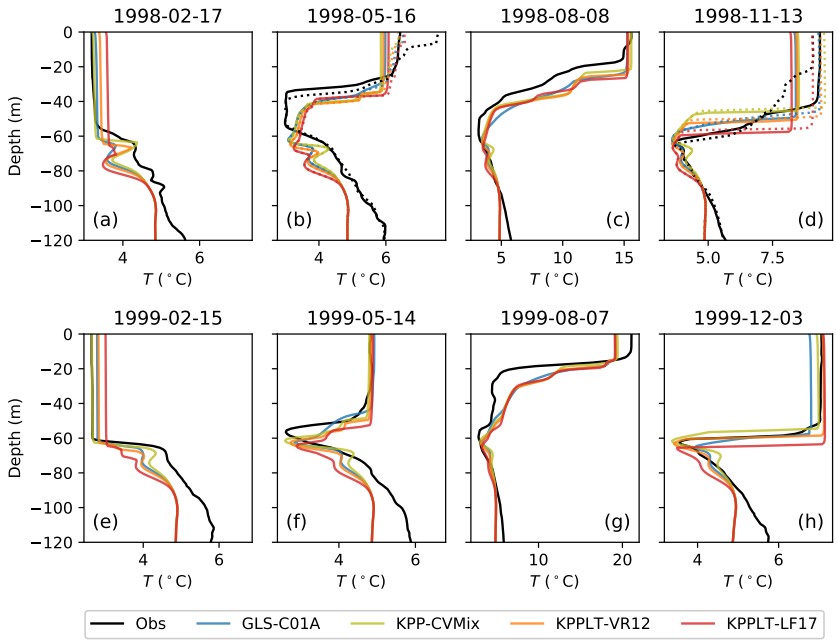

**Figure 12.** A comparison of the temperature profiles at Gotland between the observation (black) and GOTM simulations (colored). Profiles at eight time points (marked by gray diamonds in Fig. 11) are shown, representing two typical seasonal cycles. Dotted lines in panels (b) and (d) show another set of profiles one or two days apart from the solid lines (see plus signs in Fig. 11d,e), highlighting the faster time scale variations.

Fig. 13 compares the simulated temperature in the four vertical mixing schemes over the five-year period. Similar to the OSPapa case (Fig. 6), and the increased MLDs discussed in the context of Fig. 12d,h above, the most apparent differences among the four schemes are found in the phase of fall-to-winter mixed layer deepening, with KPPLT-LF17 giving the strongest deepening and KPP-CVMix the weakest. This is expected from both the design of the schemes and from results of previous tests. Overall the three KPP variants in the newly implemented CVMix module perform reasonably well in this relative long simulation.

### 3.5 Numerical sensitivity and performance

To illustrate the sensitivity of the four vertical mixing schemes to the vertical resolution and time step, we repeated the OSPapa case with combinations of different vertical resolutions of $\Delta z = [1, 5]$ m and time steps of $\Delta t = [60, 600, 1800]$ s. These configurations roughly span the range of vertical resolutions and time steps commonly used in regional and global ocean models. Fig. 14 shows the simulated temperature and MLD for KPP-CVMix, as well as the deviations from this reference run for coarser vertical resolutions and larger time steps. It is clearly seen that KPP-CVMix is very sensitive to the vertical resolution (panels (d)-(f)), but quite insensitive to the time step (panels (b), (c)). This is consistent with what has been shown in previous



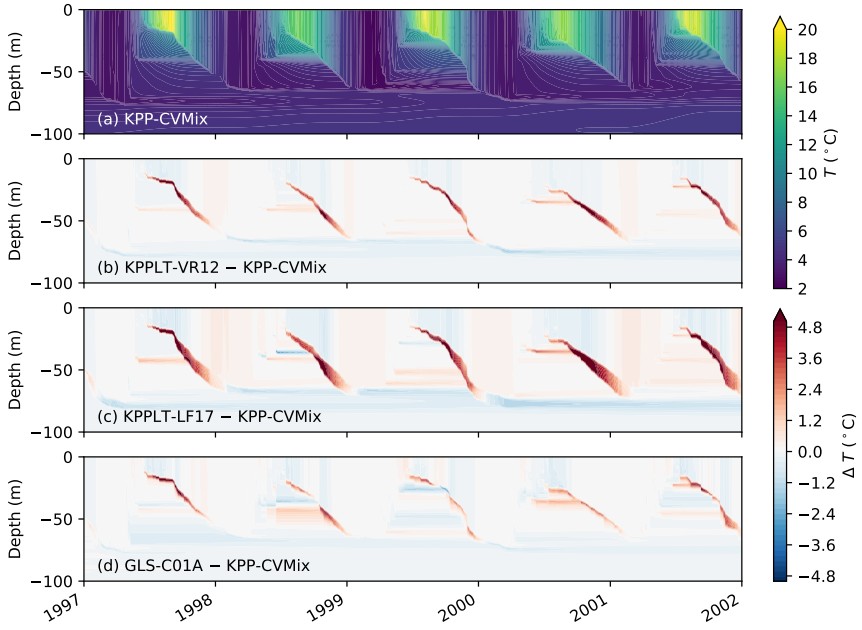

**Figure 13.** Hovmöller diagrams comparing the simulated temperature between the four vertical mixing schemes at Gotland. (a) shows the simulated temperature in KPP-CVMix. (b)-(d) show the differences from KPP-CVMix in KPPLT-VR12, KPPLT-LF17 and GLS-C01A.

studies (e.g., Van Roekel et al., 2018; Li et al., 2019). Here, we note that KPP-CVMix is most sensitive to the vertical resolution when the mixed layer is deepening (e.g., throughout phase III and sporadic events in phase II), with stronger deepening when the resolution is coarsened. This is related to the representation of mixed layer entrainment in KPP via a criterion based on the bulk Richardson number, which is particularly sensitive to the detailed choices of the numerics (Van Roekel et al., 2018). Since both KPPLT-VR12 and KPPLT-LF17 are based on KPP-CVMix, they suffer from similar sensitivity to the vertical resolution

(not shown). Note, however, that the relative effects of including Langmuir turbulence in KPPLT-VR12 and KPPLT-LF17 as shown in Fig. 6 do not change with the vertical resolution.

      In comparison, as shown in Fig. 15, GLS-C01A is much less sensitive to the vertical resolution than KPP-CVMix. Fig. 15d shows that during the phases of strong mixed layer deepening (in particular, phase III), differences in MLD and mixed-layer temperatures are approximately an order of magnitude smaller compared to KPP-CVMix if the grid spacing is increased from

1 m to 5 m. However, GLS-C01A is noticeably more sensitive to the time step than KPP-CVMix (Fig. 15b,c). This is likely related to the fact that GLS-C01A solves prognostic equations for the turbulent velocity scale and length scale, as compared to the diagnostic algorithms in KPP. However, the numerical solution of partial differential equations in GLS-C01A has the important advantage that numerical errors are guaranteed to decrease according to the well-defined convergence properties of the numerical schemes if the vertical grid spacing is reduced. For the algorithms in KPP-CVMix, this property cannot be

proved.





We also compared the execution (CPU) times in GLS-C01A and KPP-CVMix for all the relevant subroutines in each scheme, excluding common subroutines such as the mean flow equations and data input/output. Surprisingly, the CPU time required for a single time step in KPP-CVMix turned out to be 3-4 times larger than for the second-order turbulence model in GLS-C01A. It should be noted that this relative timing information only applies for the specific numerical implementations in GOTM

and CVMix used in our study. The actual performance of the corresponding second-order and KPP mixing schemes may be improved by optimizing the loop structure and the time step, which may also change the relative performance of these models. From a practical point of view, the execution times will also depend on the time step and grid spacing chosen to yield a desired accuracy. As shown above, the second-moment closures in GOTM will tolerate the use of coarser vertical grids, whereas KPP-CVMix will provide sufficient accuracy also for larger time steps. Overall, therefore, our study does not show any clear

advantage in computational costs for either model. This is a rather remarkable result as KPP-type models are generally believed to be more robust, and therefore preferable, in coarse-resolution global modeling.

## 4  Discussions and conclusions

In this paper, we documented a set of recent extensions of ocean turbulence modeling toolbox GOTM, aiming to incorporate a suite of recently developed vertical mixing schemes that include the effects of Langmuir turbulence. This new capability is

realized by adding two modules in GOTM: a CVMix module to interface with the CVMix library, and a Stokes drift module to provide the Stokes drift profiles for the Langmuir turbulence parameterizations. These modules were consistently integrated into the existing module structure of GOTM, now also allowing for flexible model setups through the newly developed YAML-based configuration lists, as well as for a straightforward integration of additional Langmuir turbulence parameterizations in the future. For example, future development of CVMix, which has been widely used in many ocean climate models, can now

be easily incorporated in GOTM through the new CVMix module, and are therefore available to other ocean models that use GOTM as their turbulence library with modest code changes. In addition, even though we have demonstrated in Section 3.2 that the two variants of KPP with Langmuir turbulence are not particularly sensitive to the details of the Stokes drift profiles (consistent with previous studies), the full Stokes drift profiles are provided by the Stokes drift module, which will facilitate future development and incorporation of Langmuir turbulence parameterizations in the GOTM framework, such as the second

moment closures of Langmuir turbulence by Harcourt (2013, 2015).

Using these two new modules in GOTM, three variants of KPP in CVMix, with and without Langmuir turbulence, were tested and compared with a second-moment turbulence closure scheme based on the GLS-framework in the $k$-$\varepsilon$ formulation. We investigated four test cases, and evaluated the model performance against available theoretical scalings and observations. These four test cases included an idealized wind-driven entrainment case and three more realistic cases: the Ocean Station

Papa, the northern North Sea, and the central Gotland Sea, each focusing on slightly different aspects of the vertical mixing processes. The results are consistent with previous studies of Langmuir turbulence effects in KPP (e.g., Li and Fox-Kemper, 2017; Li et al., 2019). Although the degree to which we can evaluate these vertical mixing schemes is still limited by the use of single column simulations, interesting conclusions can still be drawn from the direct comparison with available theoretical



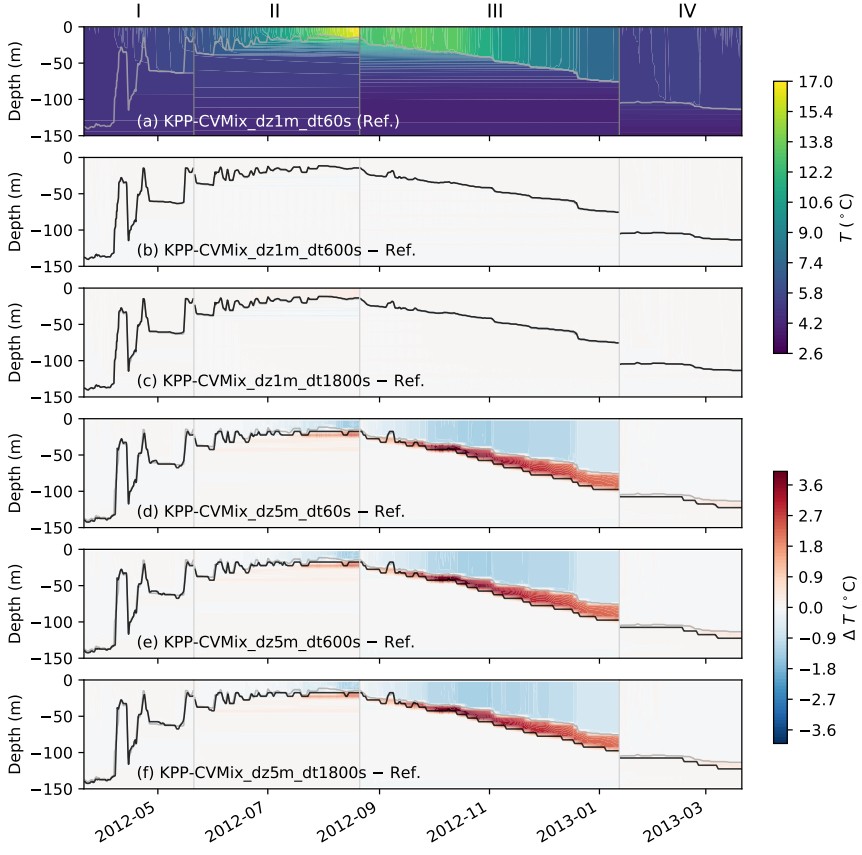

**Figure 14.** Similar to Fig. 6, but showing the differences in the simulated temperature at Ocean Station Papa in KPP-CVMix with different vertical resolution and time step. (a) shows the simulated temperature in the finest resolution and time step as in Fig. 6 (the reference). (b)-(f) show the differences from the reference with different combinations of coarser vertical resolution and larger time step.

scalings and observations, which has never been done particularly for KPPLT-VR12 and KPPLT-LF17. The effects of Langmuir
turbulence as represented in KPPLT-VR12 and KPPLT-LF17 in these test cases are most important during periods when the mixed layer is deepening. Such effects on the simulated SST, SSS and MLD in a single column setup can sometimes be comparable to the effects of the missing advection and lateral processes at Ocean Station Papa (inferred from the mismatch between modeled and observed). The magnitude of such effects also appears to be comparable to or even larger than the effects of tidal forcing in the northern North Sea in the FLEX case. These results highlight the importance of correctly representing
the effects of Langmuir turbulence in an ocean vertical mixing scheme. Compared with the $k$-$\varepsilon$ scheme, all three variants of KPP suffer from much higher sensitivity to the vertical resolution, consistent with previous studies (e.g., Van Roekel et al., 2018; Li et al., 2019), but lower sensitivity to the time step.

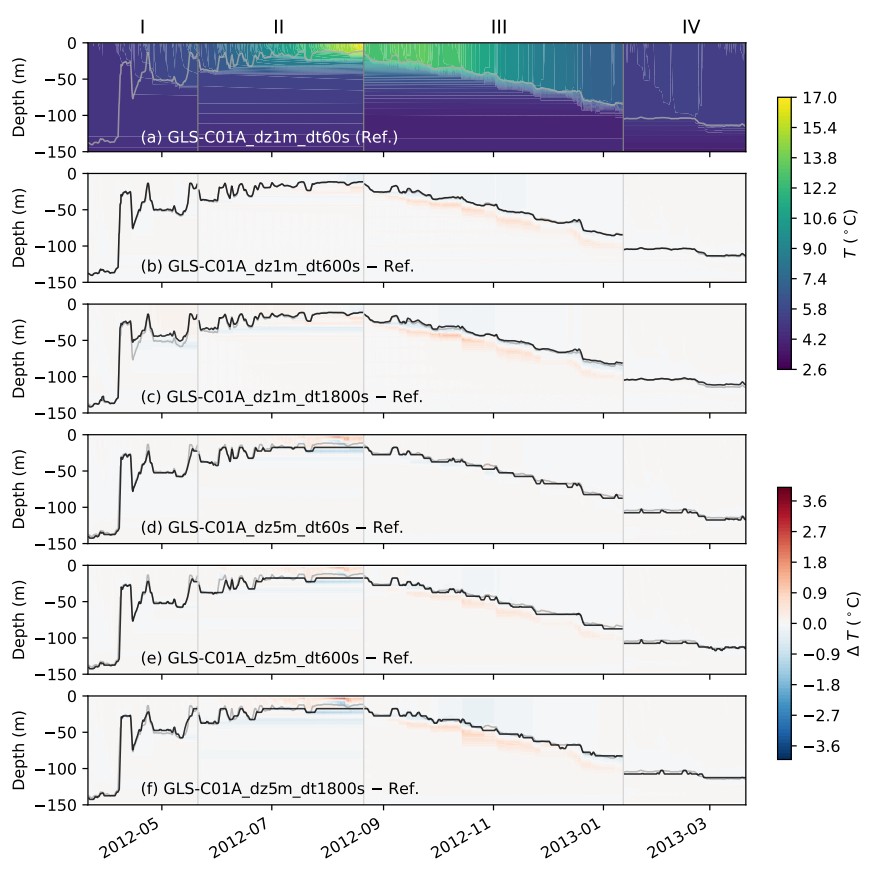

**Figure 15.** Same as Fig. 14, but for GLS-C01A.

The newly developed YAML-based configuration in GOTM also enables an easier GOTM workflow using Python and Jupyter Notebook, both gaining popularity rapidly over the years in the broader scientific community. This provides an interface
to use GOTM interactively in the Jupyter Notebook environment, especially as a single water column model which has been extremely useful in both parameterization development and evaluation (e.g., Burchard and Bolding, 2001; Burchard et al., 2008; Reichl et al., 2016; Li et al., 2019) and parameter space exploration (e.g., Li et al., 2019; Dong et al., 2020). As an example, all the simulations and figures in this paper are conducted and produced in the Jupyter Notebook environment using this new workflow. Both the source code for this new workflow and the Jupyter notebooks for this paper are available online
for maximum reproducibility (see the *Code and data availability* below for more details).

This study represents the initial steps extending the capability of GOTM to include Langmuir turbulence parameterizations, although we note that GOTM has already been used in previous studies in developing and evaluating Langmuir turbulence parameterizations (e.g., Reichl et al., 2016; Li et al., 2019). Even though only a limited set of KPP variants is included, the development here facilitates future incorporation of second moment closure schemes of Langmuir turbulence (e.g., Harcourt,

2013, 2015), as well as the development and evaluation of new Langmuir turbulence parameterizations in the GOTM framework. The Python and Jupyter Notebook based GOTM workflow enabled by the YAML-based configuration also makes future applications of GOTM easier.

It should be noted that GOTM can also be coupled to the frequently used Framework for Aquatic Biogeochemical Models (FABM, Bruggeman and Bolding, 2014) to study the evolution of marine ecosystems. The effect of different turbulence models

for ecosystem-related questions can now be evaluated within a single modeling framework. However, a systematic evaluation of such effect is beyond the scope of this work and is left for future study.

*Code and data availability.*  The source code of GOTM (v6.0) is available at https://zenodo.org/record/4541583. The Python tools for the new workflow are available at https://doi.org/10.5281/zenodo.4314038. The Jupyter notebooks and data to run the GOTM simulations and to reproduce the figures are available at https://doi.org/10.5281/zenodo.4314036.

*Author contributions.*  QL and KB implemented the CVMix and Stokes drift modules in GOTM. JB and KB developed the YAML-based configuration in GOTM. QL developed the Python tools for GOTM. QL conducted the GOTM simulations and analyses with help and guidance from HB and LU. QL led the writing of this manuscript with contributions from all authors at all stages.

*Competing interests.*  The authors declare no competing interests.

*Acknowledgements.*  We thank Ulf Gräwe at Leibniz Institute for Baltic Sea Research Warnemünde for providing the meteorological and

wave data for the Gotland Basin case. QL acknowledges the support by the Earth System Model Development program area of the U.S. Department of Energy, Office of Science, Office of Biological and Environmental Research as part of the multi-program, collaborative Integrated Coastal Modeling (ICoM) project. LU is grateful for the support by the German Research Foundation (DFG) for subproject L4 ("Energy-consistent atmosphere ocean coupling"), embedded in the Collaborative Research Centre TRR 181 "Energy Transfers in Atmosphere and Ocean". KK and HB are supported by TRR 181 as well. This project did also fund parts of the integration of CVMix into GOTM

carried out by KB and JB who recieved further support from IOW to improve the YAML-based GOTM configurations.





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
