# Peer review of "Integrating CVMix into GOTM (v6.0): A consistent framework for testing, comparing, and applying ocean mixing schemes"

_Geoscientific Model Development, 2020_

## Author Comment (AC1)

**Response to Reviewers**

We thank all the reviewers for their very helpful comments! All the comments were seriously considered and the manuscript was revised accordingly. All the revisions are highlighted in the file "Revision.pdf" by showing the differences between the revised version and the original version. In addition, the ranges of x-axes of Fig. 5e-h have been adjusted to address the second comment from Reviewer 2. A new version of the GOTM source code (v6.0.2) is provided, which fixes an issue when compiling with gfortran 11 in the previous version v6.0.0. We rerun all the test cases with this new version. Since there is no change in the physics, the results are identical with the version v6.0.0.

A point-by-point response to each reviewer is detailed in the following. All line numbers mentioned in this response correspond to the revised manuscript.

**1. Response to Reviewer 1**

I found this paper to be well-written, easy-to-read, and extremely informative. The background sections explain what experiments are being run and why, and the experiments themselves are quite thorough. Incorporating CVMix in GOTM is a novel approach to modeling vertical mixing that will hopefully be a boon to the rest of the community. The paper provides enough details for others to repeat the experiments on their own, and I agree with how the authors interpret the results.

Thank you for these comments!

The only small quibble I have is with the sentence beginning on line 184 (in section 3.1): "KPP-CVMix predicts an enhanced turbulent viscosity than GLS-C01A..."; "predicts an enhanced turbulent viscosity than" is a klunky phrasing. I'd recommend "predicts more enhanced turbulent viscosity", but there are probably other ways to clean up that sentence as well.

This sentence is revised to "KPP-CVMix predicts higher turbulent viscosity than GLS-C01A" (Line 186).

**2. Response to Reviewer 2**

The authors implemented three different KPP-based vertical mixing schemes in the GOTM, among two of which include LT-enhanced vertical mixing or entrainment, and then systematically assessed their performances by comparing to observations of three oceanic scenarios, yielding consistency with previous studies and improvements in the

LT-parameterized schemes, particularly in significant mixed layer deepening events during winter. This work provides both framework and example of implementing LT-effects into ocean models, which benefits the ocean modeling community. In general, the paper is well written and supports its conclusions as well as findings except for some minor places, which may need to be further clarified.

Thank you for these comments!

1. In section 3, table 1, the flags for Ekman depth limit and MO length limit are both false while there exist strong heating events (re-stratification in FLEX) and significant mixed layer deepening (Gotland Basin). Can you clarify why these two depth limits are not considered here?

These choices to have both the Ekman depth limit and MO length limit off are recommended by the CVMix documentation (Griffies et al., 2015) as there is little sensitivity with their use, and therefore they are the default settings in CVMix. These default settings in Table 1 are based on the NCAR implementation of KPP, and they are commonly used. We rerun the FLEX and Gotland Basin cases with both limits turned on and the differences in the results are indeed very small. Since our focus is not on the sensitivity of each parameterization scheme to the different choices of parameters, we choose to use the default settings for CVMix. We note, however, that the framework described here makes it very easy to explore such sensitivity to the different choices of parameters. We added a note in the text to clarify this when describing the parameters in Table 1 (Lines 156-159).

2. In your Fig.5, panel IV, why the temperature bias in the well-mixed upper layer is much greater than the salinity bias in the same layer? It seems the salinity prediction is generally better than temperature. Can you briefly explain why?

This impression of smaller salinity bias is probably an artifact due to the choice of the relatively bigger range for the x-axis in Fig. 5h (in order to show the strong salinity stratification at depth). This can be seen by comparing panels (e) and (f) of Fig. 4. Compared to their respective seasonal variation, the simulated SST often has smaller bias than the simulated SSS (see, e.g., stage IV). Therefore, the difference between the simulated and observed temperature in Fig. 5d is actually smaller than the difference between the simulated and observed salinity in Fig. 5h, when compared to their seasonal variations, respectively. To address this, the x-axes in panels (e)-(h) of Fig. 5 are now adjusted to focus more on the differences in the mixed layer.

3. Section 3.2, Page 12, do you have any discussion on wind-wave misalignment effects on LT intensity and thus their effects on turbulent mixing?

The effects of wind-wave misalignment on LT intensity and therefore on turbulent mixing are extensively studied in Van Roekel et al., 2012 using large eddy simulations and discussed

in Li et al., 2016 in the context of parameterization. These studies are the basis of KPPLT-VR12. However, we feel that this particular effect is beyond the scope of this study. And the GOTM simulations with four vertical mixing schemes in this study are not sufficient for us to discuss the effects of wind-wave misalignment in detail. One may set up sensitivity tests using this framework and use the Ocean Station Papa case to check the effects of wind-wave misalignment in KPPLT-VR12. But such effects are limited to what is parameterized in KPPLT-VR12. A better way is probably to use large eddy simulations such as those in Van Roekel et al., 2012 but with realistic forcing conditions at Ocean Station Papa. This is now clarified in Lines 227-230.

4. Page 16, L335-336, Can you be more specific about why a second-order turbulence closure scheme will be more helpful?

The primary reason is that the current implementation of KPP in CVMix does not contain a bottom boundary layer. Even if we had implemented a bottom boundary layer in KPP, a special treatment would have been needed for cases of merging surface and bottom boundary layers (e.g., Durski et al., 2004), as KPP treats boundary layer mixing and interior mixing differently. A second-order turbulence closure is more natural in dealing with these scenarios. And Langmuir turbulence parameterizations based on second-order turbulence closures such as Harcourt 2013; 2015 will likely be helpful to quantify the combined effects of Langmuir turbulence and tides. We have added a note to clarify this (Lines 342-345).

5. Page 19, L371-375, small Langmuir number is also occasionally observed in the spring or summer, can you explain why LT effects are most pronounced in the winter? Is it easier for LT to erode halocline than thermocline?

In the Gotland Basin case, the salinity is nudged to the observation with a time scale of 50 days (mentioned on Lines 356-358). This is the reason why the discussion of this test case is focused on the temperature. Essentially there should be no difference between eroding halocline and eroding thermocline by LT -- at least in these LT parameterizations where it is the buoyancy that matters. The reason to discuss the halocline erosion here is that, as shown in Fig. 12, the temperature stratification below ~60 m is unstable -- it is the halocline that maintains the stable stratification below the winter mixed layer.

The LT effects seem more pronounced in winter mainly for two reasons. (1) the seasonal thermocline is shallow and strong in summer -- the enhanced mixing by LT has a smaller impact on deepening the mixed layer depth in summer, even though the impact on the relative deepening in percentage may sometimes be bigger; (2) in KPPLT-LF17 the LT induced entrainment at the base of the mixed layer is only applied when the surface buoyancy forcing is unstable, as it is still unclear how LT affects the entrainment in stable conditions (see more discussions of this in Li and Fox-Kemper, 2017).

We have added some notes to clarify these points (Lines 383-385 and Lines 387-388).

**References**

Durski, S. M., Glenn, S. M., & Haidvogel, D. B. (2004). Vertical mixing schemes in the coastal ocean: Comparison of the level 2.5 Mellor-Yamada scheme with an enhanced version of the K profile parameterization. Journal of Geophysical Research: Oceans, 109(C1). https://doi.org/10.1029/2002JC001702

Griffies, S. M., Levy, M., Adcroft, A. J., Danabasoglu, G., Hallberg, R. W., Jacobsen, D., et al. (2015). Theory and numerics of the Community Ocean Vertical Mixing (CVMix) project (pp. 1–98).

Harcourt, R. R. (2013). A second-moment closure model of Langmuir turbulence. Journal of Physical Oceanography, 43(4), 673–697. https://doi.org/10.1175/JPO-D-12-0105.1

Harcourt, R. R. (2015). An improved second-moment closure model of Langmuir turbulence. Journal of Physical Oceanography, 45(4), 84–103. https://doi.org/10.1175/JPO-D-14-0046.1

Li, Q., & Fox-Kemper, B. (2017). Assessing the effects of Langmuir turbulence on the entrainment buoyancy flux in the ocean surface boundary layer. Journal of Physical Oceanography, 47(12), 2863–2886. https://doi.org/10.1175/JPO-D-17-0085.1

Li, Q., Webb, A., Fox-Kemper, B., Craig, A., Danabasoglu, G., Large, W. G., & Vertenstein, M. (2016). Langmuir mixing effects on global climate: WAVEWATCH III in CESM. Ocean Modelling, 103, 145–160. https://doi.org/10.1016/j.ocemod.2015.07.020

Van Roekel, L., Fox-Kemper, B., Sullivan, P. P., Hamlington, P. E., & Haney, S. R. (2012). The form and orientation of Langmuir cells for misaligned winds and waves. Journal of Geophysical Research, 117(C05001), C05001. https://doi.org/10.1029/2011JC007516